# Toward Improving Diagnostic Strategies in Chronic Disorders of Consciousness: An Overview on the (Re-)Emergent Role of Neurophysiology

**DOI:** 10.3390/brainsci10010042

**Published:** 2020-01-10

**Authors:** Luana Billeri, Serena Filoni, Emanuele Francesco Russo, Simona Portaro, David Militi, Rocco Salvatore Calabrò, Antonino Naro

**Affiliations:** 1IRCCS Centro Neurolesi Bonino Pulejo, 98124 Messina, Italy; luana.billeri@irccsme.it (L.B.); simona.portaro@irccsme.it (S.P.); g.naro11@alice.it (A.N.); 2Padre Pio Foundation and Rehabilitation Centers, San Giovanni Rotondo, 71013 Foggia, Italy; uo.tecnologica@fondazionepadrepio-onlus.it; 3Stomatodental Center, 98100 Messina, Italy; david.militi@alice.it

**Keywords:** disorder of consciousness (DoC), unresponsive wakefulness syndrome (UWS), minimally conscious state (MCS), functional Locked-In Syndrome (fLIS), resting state, passive paradigms, neurophysiology

## Abstract

The differential diagnosis of patients with Disorder of Consciousness (DoC), in particular in the chronic phase, is significantly difficult. Actually, about 40% of patients with unresponsive wakefulness syndrome (UWS) and the minimally conscious state (MCS) are misdiagnosed. Indeed, only advanced paraclinical approaches, including advanced EEG analyses, can allow achieving a more reliable diagnosis, that is, discovering residual traces of awareness in patients with UWS (namely, functional Locked-In Syndrome (fLIS)). These approaches aim at capturing the residual brain network models, at rest or that may be activated in response to relevant stimuli, which may be appropriate for awareness to emerge (despite their insufficiency to generate purposeful motor behaviors). For this, different brain network models have been studied in patients with DoC by using sensory stimuli (i.e., passive tasks), probing response to commands (i.e., active tasks), and during resting-state. Since it can be difficult for patients with DoC to perform even simple active tasks, this scoping review aims at summarizing the current, innovative neurophysiological examination methods in resting state/passive modality to differentiate and prognosticate patients with DoC. We conclude that the electrophysiologically-based diagnostic procedures represent an important resource for diagnosis, prognosis, and, therefore, management of patients with DoC, using advance passive and resting state paradigm analyses for the patients who lie in the “greyzones” between MCS, UWS, and fLIS.

## 1. Introduction

Consciousness is a multi-faceted concept definable in its simplest form as “sentience or awareness of internal or external existence” [1]. Consciousness comprises two constituents: awareness (i.e., the content of consciousness) of the self and the environment and wakefulness (i.e., the level of consciousness). While wakefulness refers to brain general activity, awareness is a dynamic and complex process involving large-scale frontoparietal networks and cortico-thalamo-cortical loops [2]. In particular, awareness and behavioral responsiveness both depend on the functionality of large-scale cortico-thalamo-cortical networks, in addition to the ascending activator system sited in the brainstem, which oversees wakefulness generation and maintenance [2,3]. 

Disorder of Consciousness (DoC) includes cognitive dysfunction (e.g., delirium), coma, Unresponsive Wakefulness Syndrome (UWS; formerly vegetative state), Minimally Conscious State (MCS), Locked-In Syndrome (LIS), and brain death. The different degree of cortico-thalamo-cortical neuromatrix disconnection has been postulated to account for the severity and completeness of unawareness [4,5]. 

The clinical identification of awareness includes stimuli responsiveness and the observation of spontaneous behaviors. The lack of behavioral responsiveness characterizes patients with UWS, whereas inconsistent but clearly distinguishable behavioral evidence of consciousness characterizes patients with MCS [4,6]. A thin transition zone exists between pathologically altered and normal consciousness. This is defined as emerging from a minimally conscious state (eMCS), when communication skills, the use of objects, or both are regained. 

The distinction between UWS and MCS can however be subtle in many patients. In fact, clinical assessment does not always adequately detect conscious awareness. In fact, about 40% of patients with UWS can be misdiagnosed, as they may be conscious but are unable to show any signs of consciousness. These patients have been variously labeled as non-behavioral MCS, cognitive-motor dissociation, or functional Locked-In Syndrome (fLIS) [7,8,9,10,11]. fLIS represents a pathophysiologically similar but behaviorally different condition from UWS. Such patients are characterized by an MCS-like pathophysiological profile and a UWS-like behavioral profile.

The relatively limited sensitivity of clinical scales to detect and rate the behavioral responsiveness in relation to awareness presence depends on the fact that such an evaluation is based on motor reactivity purely. Indeed, the lack of behavioral responsiveness does not necessarily mean unawareness, because patients may have motoric and cognitive deficits biasing clinical evaluation [12]. Therefore, a differential diagnosis can be achieved only through paraclinical approaches [7]. These approaches aim at capturing the residual brain network models, at rest or that may be activated in response to relevant stimuli, which may be appropriate for awareness to emerge (despite their insufficiency to generate purposeful motor behaviors) [13]. In this regard, different brain network models have been studied in patients with DoC by using sensory stimuli (i.e., passive tasks), probing response to commands (i.e., active tasks), and during resting-state [13,14]. 

Response to commands is investigated as a signature of residual, covert awareness. Various tasks are employable to this end, including the vision of moving colored shapes, familiar faces, bright and flashes; listening to a simple click, noise, words alone, meaningful speech, meaningless or ambiguous speeches, and one’s name; and the perception of pain. Brain activity is recorded in terms of metabolic (Positron Emission Tomography (PET)), hemodynamic (functional Magnetic Resonance Imaging (fMRI)), and brain connectivity variations (fMRI, electroencephalogram (EEG), and magnetoencephalography (MEG)) during task performing [15,16,17,18,19]. The information gained by such tasks is twofold. First, we can understand which areas may be involved in the sensorimotor elaboration of a stimulus, and up to what level of complexity it is processed, without the patient having to actively participate in the task. Second, the objective signs of any form of voluntary communication can be evaluated as the patient is required to collaborate. Overall, a positive response to the active task suggests awareness in a patient. On the other hand, a negative response to the active task does not necessarily indicate unawareness, as there are many factors explaining such a negative outcome despite a residual awareness including a very slight response to stimuli and their short-term and fluctuable nature [20]. 

The evaluation of resting state may be particularly appropriate for patients with DoC, owing to their possibly limited collaboration skills to an active task. Neuroimaging evaluations suggest a specific brain organization during the resting state, which includes a network within the posterior cingulate cortex, the adjacent precuneus, the anterior cingulate cortex, and frontal regions, known as the default mode network (DMN). Normally, this network is negatively correlated with a system within lateral fronto-parietal areas, known as task-positive network. Both these networks subtend thoughts and perception of the environment [21,22,23,24,25]. The functional correlation between these networks is metabolically preserved in patients with MCS but not in those with UWS [26]. Thus, a between-network anti-correlation could be a distinctive feature of MCS [27,28,29,30,31].

In the passive paradigms, cerebral activation (estimated by metabolic, PET, hemodynamic, fMRI, and brain connectivity variations (fMRI, EEG, evoked potentials (EPs), Event-Related potentials (ERPs), and MEG)) are achieved thanks to external stimuli that do not require the patient to actively participate to the task. Generally, a stimulus activates only the primary “typical” areas (somatosensory, auditory, and visual) in patients in UWS, while the same stimuli involve both primary ones and high-order secondary association areas in patients in MCS, eMCS, LIS, and fLIS and in healthy controls [19,32,33,34,35,36]. However, the identification of stimulus induced cerebral activation does not necessarily indicate awareness, even when high-order secondary association areas are activated by a stimulus [13].

Therefore, each of the paradigms has strengths and weaknesses concerning DoC differential diagnosis, with particular regard to fLIS. The main problem with this DoC entity is likely related to its pathophysiology. Patients suffering from fLIS show an extreme impairment in motor behavior despite a partial preservation of higher cognitive functions [7,37,38,39]. This dissociation is likely to result from the injury of one or more levels of the sensorimotor system rather than to brain connectivity breakdown [37,38,39]. fLIS has to be differentiated from “classic” LIS (i.e., quadriplegia and anarthria, with some spared eye movements) and “total” LIS (complete immobility). Noteworthy, patients with LIS suffer from a lesion of the brainstem rather than of the diencephalon and brain hemispheres. Therefore, they clinically appear as unaware (owing to the severe sensory-motor deficit), whereas self-awareness and cognitive abilities are preserved, as well as a normal network connectivity of the brain [40]. 

Since performing even simple tasks can be difficult for patients with DoC, our and other research groups made many efforts for identifying putative awareness markers in both resting-state and passive paradigms, mainly adopting advanced EEG/ERPs analyses. Herein, we propose a scoping review aimed at highlighting the (re-)emergent role of neurophysiology in terms of current, innovative neurophysiological examination methods in resting state/passive modality to improve the diagnostic strategies in chronic DoC in support of the irreplaceable value of bedside, prolonged, clinical examination.

## 2. Literature Review

### 2.1. Methods

We carried out a scoping review (that identifies the nature and extent of research evidence, including the ongoing research) of the innovative neurophysiological examination methods in resting state/passive modality created for patient with DoC, with particular regard to those suspected to be behaviorally nonresponsive and covertly aware. The present study was approved by the Local Ethics Board. The search on PubMed/MEDLINE, ScienceDirect, Scopus, and Google Scholar identified the already existing neurophysiological paradigms designed to be administered in patient with DoC and used by researchers to evaluate neurophysiologic processes in patients with severe brain damage or neurodegenerative diseases leading to behavioral unresponsiveness and covert awareness. We used the following keywords: (1) disorder of consciousness AND cover awareness AND neurophysiology (#1); (2) unresponsive wakefulness syndrome OR minimally conscious state OR locked-in syndrome OR resting state OR passive paradigm (#2); and (3) #1 AND #2. Papers were screened on the basis of title, title/abstract, and full-text review. The innovative neurophysiological examination method needed to be designed for patients with DoC, include resting state/passive modality, and be aimed at improving the diagnostic/prognostic strategies in adult, chronic DoC. The search identified a total of 426 studies to be screened, of which 120 were included in the review.

### 2.2. Electroencephalogram Assessment

EEG oscillation changes reflect the temporal synchronization of pyramidal cortical neurons by the summation of the postsynaptic potentials to the neuronal dendrites. This arises from the output of the ascending activator system and the thalamo-cortical loops. The alpha (approximately 8–12 Hz) and theta (approximately 4–7 Hz) frequencies are markers of human cognition processes and conscious awareness [41], and are both abnormal in DoC [42,43]. Notably, there is a greater relative power in the delta band in patients with UWS as compared to those with MCS [44,45]. In MCS, EEG appears to show widespread slowing of brain activity, although reactive to external stimuli [44]. Furthermore, the site and type of brain lesions significantly affect EEG abnormalities, including slowing, theta and delta frequency disorganization, and reactivity of the alpha rhythm [44]. Clinical recovery from a UWS can be associated to a decrease in delta and theta activity and the reappearance of a reactive alpha rhythm. Further, the cortical sources are important concerning both differential diagnosis and prognosis [46,47]. It is also possible that EEG reactivity to sensory stimuli (at least one type of) may allow distinguishing patients with MCS from UWS [48].

Even though EEG is a very available and easily applicable tool to patients with DoC, the available data are however neither sufficient to establish a typical model nor relevant to disclose awareness, even covert [42,49,50,51]. In this regard, the advanced analysis of EEG recorded in the resting state or induced by brain stimulation paradigms (including spectral analysis, complexity measures, microstate analysis, approximate entropy, cross-approximate entropy, permutation entropy, spectral power, spectral entropy, coherence, imaginary coherence, symbolic transfer entropy, weighted symbolic mutual information, debiased weighted phase lag index, complex network analyses, phase lag index, Lempel-Ziv complexity, bispectral index, predominant background activity, wavelet decomposition, and intrinsic network reactivity index) can provide an appropriate approach to differentiate patients with DoC, in addition to clinical evaluation at a group level [45,50,51,52,53,54,55,56,57,58,59,60,61,62,63,64,65,66,67,68,69,70,71,72,73,74,75], with also some insight at individual level [63,69,76,77,78].

This is a more advanced step in the differential diagnosis, as it requires denser EEG recording electrodes (e.g., [63,64,73,74]) and specific analytic and statistical procedures. In particular, EEG-based connectivity assessment is useful to differentiate and prognosticate patients with DoC, especially using active tasks [79]. We found that specific passive paradigms could allow the identification of a residual functional connectivity (FC) in the fronto-parietal networks in the θ and γ frequency bands [47], which are important biomarkers of ongoing, even hidden, cognitive processing of sensorimotor information. Going further, it is known that awareness may depend on the highly dynamic FC of large-scale cortico-thalamo-cortical networks [80]. The analysis of dynamic FC (DFC) (i.e., brain connectivity changes over time) in the resting state showed DFC variations in terms of matrices and topography (above all in γ frequency band) in patients with MCS but bot in those with UWS. In particular, there was a significant correlation between DFC magnitude and the level of behavioral responsiveness as measured by the Coma Recovery Scale-Revised. Thus, DFC analysis seems promising to DoC differential diagnosis at a group level. Moreover, DFC analysis may increase the current knowledge of the awareness processes, thus helping to design neuromodulation protocols aimed at regaining awareness by targeting maladaptive and dysfunctional FC.

### 2.3. Evoked Potentials

Both EPs and ERPs test the integrity of the central nervous system and assess its perceptual and cognitive processing reactivity in patients with DoC, thus trying to provide information on diagnosis and prognosis at individual and group level [56]. In particular, the deterioration of the cortical response at sensory evoked potentials is indicative of poor prognosis, and the absence of N20 is related to a poorer prognosis in comatose patients [59]. Brainstem auditory evoked potentials can be either normal or with a deteriorated waveform morphology [59]. Overall, short-latency EPs assessment is very available and of easy application and clinical interpretability, but it is poorly informative on functional and effective cortico-cortical connectivity [79], which is essential to capture the brain network dynamic subtending awareness generation and maintenance. Therefore, advanced analysis of EEG signals induced by specific stimuli (i.e., long-latency EPs -ERPs) are mandatory. Even though some ERPs studies do not show any significant difference at a group level by using sensory stimuli [81,82,83], specific ERPs assessments can provide partial information on cortico-cortical connectivity and, consequently, they are of moderate diagnostic and prognostic utility, especially in identifying fLIS among UWS (i.e., low sensitivity but high specificity) [76,84,85,86]. Similar to short-latency EPs assessment, ERPs methodology is very available and of easy clinical interpretability, but its application requires that the visual inspection of ERPs has to be integrated with statistical analysis methods [76]. ERPs are also useful indicators of chances of recovery of consciousness [49], including echoic memory (MMN), acoustical and semantic discrimination (P300), and incongruent language detection (N400), as well as in differentiating between UWS and MCS. However, P300 is not a reliable marker of awareness [75,81,83,84,87,88], even though a partially preserved semantic processing can be observed when detecting salient stimuli, such as the subject’s own name [89].

We expanded the role of ERPs in residual awareness detection by describing the effects of Non-Invasive Brain Stimulation (NIBS) (including transcranial magnetic (TMS) and direct/alternating current stimulation (tDCS and tACS, respectively)) on the spectral properties of ERPs elicited by visual and auditory stimulation (according to an acoustical and/or visual discrimination paradigm (P300)) in patients with DoC. The rationale of assessing the spectral content of ERPs with regard to the missing diagnosis of fLIS relied on the fact that specific ERPs features express the underling network processes that generate ERPs [90,91,92,93,94]. Even though these networks are not necessarily relevant for awareness to emerge, spectrum power, coherence, entropy spectrum, non-linear analysis, information theory, and functional connectivity were the most useful features to differentiate consciousness levels, predict outcome, and measure patients’ brain responses to brain interventions [62,76,95]. 

Furthermore, the study of large-scale sensory-motor integration (SMI) processes subtending awareness can be useful to identify patients with fLIS [89,96,97]. The usefulness of studying SMI processes concerning awareness detection relies on the fact that SMI depends on large cortical-subcortical networks carrying information at a conscious level [98,99]. Instead, first-level SMI processes (i.e., within primary sensorimotor cortices) do not correlate with awareness [100,101]. Interestingly, we found that the evaluation and modulation of SMI through NIBS was useful in UWS–fLIS differential diagnosis. We evaluated functional visuo-premotor-motor FC in UWS and MCS sample before and after the administration of an associative stimulation protocol composed of tDCS on the dorsolateral prefrontal cortex and parieto-occipital areas and trans-orbital alternating current stimulation [96,97]. Such an approach confirmed the clinically-based differential diagnosis with the exception of one patient in UWS who electrophysiologically behaved as those in MCS, and who regained consciousness some months later. Therefore, passive audiomotor and visuomotor integration could be a promising tool to stimulate sensory networks FC, thus enabling physicians in the differential diagnosis among MCS, UWS, and fLIS [96,97]. Based on our studies, we can argue that patients UWS showing an improvement in cortical FC and motor output after a SMI-based NIBS approach should be considered as suffering from fLIS rather than UWS. 

### 2.4. Pain

The appropriate treatment of pain in patients with DoC is extremely challenging [102]. In fact, the perception of pain is difficult to be evaluated in patients with UWS, who show unconscious reflexive behaviors. Pain assessment can also be difficult in patients with MCS, who show conscious and reproducible behaviors, which can be however fluctuating [103]. The old concept that patients with UWS do not feel pain has been totally abandoned thanks to the evidences of aware pain perception coming from fMRI and laser-evoked potentials (LEPs) studies and by the assessment of cognitive processes related to the autonomic responses to nociceptive stimuli [104,105,106]. In this regard, the presence of N2P2 components of Aδ-LEPs and CLEPs permitted to differentiate MCS from UWS [107].

LEPs are commonly used in the study of nociceptive pathways. However, their use requires the following precautionary considerations: (i) basic LEPs parameters (i.e., amplitude and latency) cannot provide unequivocal information on pain perception in patients with DoC [104]; (ii) LEPs are mainly a stimulus-dependent relevant excitation marker [108]; (iii) LEPs require the selective activation of thermo-nociceptive afferents; and (iv) LEPs are generated within different cortical areas processing both nociceptive and non-nociceptive inputs [109]. However, LEPs are somewhat useful in estimating if a patient has the potential to feel pain. In this regard, some correlations have been documented between the intensity of perceived pain and the characteristics of the single LEPs [110,111]. 

In an attempt to identify other autonomic nervous system markers differentiating DoC individuals, we quantified specific features of ultra-late laser-evoked potentials (CLEPs) and of the skin reflex (SR) (including amplitude, latency, and γ-band power (γPOW)), and 24-h-polygraphy cardiopulmonary parameters. All such measures are related to specific aspects of the cognitive processes associated to the Autonomic Nervous System (ANS) functions [112,113]. In particular, MCS showed physiological variations of O_2_ saturation, heart rate, and heart rate variability throughout the night, and a preserved SR-γPOW. On the other hand, UWS did not show significant changes. Following laser stimulation, MCS patients showed a clear increase in CLEPs-γPOW, O_2_ saturation, heart rate, and heart rate variability. Instead, UWS individuals did not show significant change, with the exception of two patients. Therefore, we argue that an extensive neurophysiological assessment of ANS may corroborate DoC differential diagnosis. The abovementioned two UWS patients showed changes in vital signs similar to MCS and sensitivity to electrophysiological pain. It is therefore conceivable that ANS assessment could be useful to identify cognitive processes related to the residual and conscious autonomous system even in some UWS patients. 

Notwithstanding, further studies on LEPs have to be promoted to furnish clinicians and researchers with a more reliable estimation of pain perception in DoC patients and achieve a differential diagnosis. For example, we found that the Gamma-Band Oscillation associated with LEPs expressed the modulation of the intensity of pain within the primary sensory area [114,115]. This modulation reflects the integrity of a network within frontal, cingulate, and parietal areas involved in pain perception and pain-gating processes, which represents a conscious process regardless of the patient’s ability to communicate [116]. Therefore, assessing specific, spectral features of LEPs may result useful for DoC differential diagnosis, even at individual level.

### 2.5. Sleep

In the last decade, many studies supported the relevant role of sleep evaluation to differentiate the UWS from individuals with MCS and to plan further rehabilitative work-up in chronic DoC patients. Generally, patients with MCS show a more conserved sleep structure (that probably reflects at least a partial functional integrity of the brain), with the presence of physiological hypnic figures and a partially reserved distribution of rapid eye movement (REM) and non-REM (NREM) sleep as compared to subjects in UWS [117]. UWS patients usually lack of the key EEG changes across the different phases of sleep (e.g., REM, sleep spindles, and vertex waves). Some of the patients with UWS show slow waves activities immediately after falling asleep, which persist or gradually increase during the sleep period. On the other hand, other UWS patients show no sleep fluctuations [118]. The deterioration of sleep–wake cycle is an indicator of brainstem dysfunction and patient’s prognosis [119,120]. In particular, a study reported that UWS patients with even severe impairment showed residual sleep patterns (including, sometimes, REM sleep), and that the presence of slow-wave sleep was correlated with high Coma Recovery Scale-Revised scores [121]. Similarly, other studies showed a discrimination between DoC categories according to sleep findings [122,123,124,125]. Furthermore, “regular” sleep structure was demonstrated as a good predictor of clinical outcome in sub-acute DoC patients and it was stronger than other known prognostic factors of DoC outcome [126,127]. These works only highlight that simple and (relatively) easy visual-quantitative sleep analysis is helpful in the prognostic and/or diagnostic evaluation of DOCs. Last, we found a correlation among global brain connectivity, sleep structure, and pain perception, which are related to the activity of the wide thalamo-cortical and cortico-cortical networks underlying consciousness [128]. In this regard, the loss of brain complexity after severe injuries was shown to depend on a pathological tendency of cortical circuits to fall into silence (OFF-period) upon receiving an input (a behavior typically observed during sleep) [129]. Therefore, matching pain and sleep data may improve DoC differential diagnosis [128]. 

### 2.6. Non-Invasive Brain Stimulation 

NIBS, delivered by using TMS and tDCS/tACS, allows targeting the neural circuitries potentially subtending awareness in the attempt to regain consciousness [130]. On the other hand, single, specific NIBS approaches can provide useful information on the neural correlates of consciousness, allow for accurate diagnosis and prognosis of consciousness disorders (which we will focus on), and provide valuable support for designing individually adapted clinical and neurorehabilitation management. The rationale of using NIBS to highlight neural patterns potentially subtending awareness relies on the fact that awareness recovery requires that neuroplasticity mechanisms reach a determined threshold for awareness to emerge [131]. This issue was well evidenced by several experimental approaches, including the assessment of EEG oscillatory microstate occurrence (fast alpha and theta/delta), and brain operational architectonics (size, instability, speed of growth, and life span of neuronal assemblies; number and strength of connections; and strength of the DMN synchrony) [132,133,134,135,136,137,138,139,140,141]. Indeed, NIBS can furnish the required, still missing amount of neuroplasticity allowing awareness to emerge, even transiently [37,76,77,78]. In keeping with this issue, it was shown that NIBS can unmask patterns of residual, covert connectivity in some UWS patients, thus helping the differential diagnosis between UWS and fLIS [137,139,140]. In particular, we found that high-frequency networks were characterized by a large-scale connectivity breakdown and a local hyperconnectivity, whereas low-frequency networks showed a partially preserved large-scale connectivity. Moreover, the FC of DMN and external awareness networks (EAN) were clearly anti-correlated [142]. Hence, we hypothesized that the patients with UWS who showed an, albeit residual, DMN-EAN anticorrelation had to be labeled as fLIS. In fact, such residual network anticorrelation is the prerequisite for cognition (and thus awareness), as well as aware pain perception [143]. Overall, the data on NIBS suggest that patients responsive to a paradigm were more likely to regain consciousness in the next months as compared to the non-responsive patients [130,144].

Notably, integrating EEG with NIBS can provide even more useful information concerning diagnosis as compared to both EPs and ERPs [85,86,132,133,142,143,145]. Indeed, NIBS-EEG paradigms provide useful connectivity and integration metrics characterized by high sensitivity and specificity for awareness detection at both individual and group level [79,146,147,148,149]. Generally, the TMS-evoked activities in UWS patients are characterized by simple and short-lasting response patterns [149,150], whereas those in MCS (and fLIS) patients are more complex [149,150,151]. Among these, the perturbational complexity index (PCI), an index of complexity of the responses of brain activity to magnetic pulses delivered through TMS, provides useful information to label DoC patients at individual level [147]. Indeed, a PCI > 0.31 is the threshold to differentiate MCS from UWS patients [145]. As limitation, this type of procedure requires high expertise and specific devices (e.g., TMS-compatible EEG amplifiers are required for recording TMS-EEG responses).

Even though the cerebellum is not directly linked to awareness generation and maintenance, the demonstration of preserved cerebellum–cerebrum interaction can be useful to demonstrate awareness preservation in patients with DoC. In fact, the cerebellum entertains several networks with the cerebral cortex through complex cortico-basal ganglia loops. The preservation of such networks is crucial to awareness to emerge and recover. In this regard, we showed that the modulation of cerebellum–cerebrum connectivity by using tDCS can be a useful approach to differentiate DoC patients at a group level [145]. Further, we also demonstrated that cerebellar tACS can induce motor adaptation during walking and hand motoric tasks that are awarely perceived by the healthy subject [152]. Therefore, cortical connectivity modulation induced by cerebellar stimulation could disclose covert cognition in patients with chronic DoC [145].

## 3. Authors’ Point of View

The differential diagnosis of DoC is still challenging, especially concerning the chronic state. Actually, the bedside clinical examination still represents the gold standard for achieving diagnosis and prognosis of DoC patients. However, electrophysiological techniques growingly play a major role in the diagnostic procedure, as they can discover (passive and resting state paradigms) or bring to light (NIBS-based paradigms) residual connectivity patterns subtending cognitive processes, i.e., covert awareness, which cannot be clinically demonstrated. Table 1 shows the different assessments we reviewed with their main outcomes concerning significant MCS/UWS differentiation at group and individual level (i.e., disclosing patients with fLIS among those with UWS and MCS vs. UWS).

Whether the clinical assessment remains fundamental and irreplaceable, the combinations of behavioral and electrophysiological/neuroimaging procedures represents an important, cautionary condition to reduce misdiagnosis [153,154,155,156,157,158,159]. Further electrophysiological studies are however required to confirm the values of basic and advanced neurophysiologic paradigms in DoC diagnosis, given that the moderate evidence concerning the possibility distinguishing MCS from UWS is in favor of a spared EEG reactivity, the presence of LEPs components, EPs features, and the PCI threshold [160]. Moreover, there is still insufficient evidence concerning resting state EEG analyses, specific resting and NIBS-based EEG connectivity measures, and cognitive ERPs [160].

Given the specific pros and cons of the single diagnostic procedures, a systematic approach combining multimodal assessment techniques and different stimulation modalities may be the most appropriate strategy to detect residual signs of consciousness as compared to single interventions. It is our opinion that two different neurophysiological approaches may be profitably enhanced in the study of patients with DoC. The first level, strictly clinical, simple and including very consistent neurophysiological testing (e.g., the presence of cortical responses to SEPs, P300 and MMN to ERPs, and the background activity of EEG both in wakefulness and in sleep) should be used in intensive care units and primary rehabilitation settings to estimate diagnosis and prognosis. The second level, mainly for research purposes or, possibly, as a second diagnostic step, including advanced and innovative neurophysiological investigations (requiring expertise, dedicated personnel, and not negligible economical resources), could be used in more specialized rehab institutes. 

It is true that these advanced approaches, despite their experimental nature, offer useful information concerning DoC differential diagnosis and fLIS identification (as confirmed by the clinical follow-up in the same studied populations). However, these approaches should be dedicated to restricted and selected groups of DoC patients when the diagnosis/prognosis based on the clinical data and the abovementioned first-level diagnostic tools is uncertain (i.e., patients at the boundary of MCS, UWS, and fLIS). This is mainly because all the recent studies, while being very interesting and very important for clarifying the pathophysiology and the neuroplasticity process subtending (un)awareness, are virtually not implementable in small centers. In fact, such approaches require the presence of well-structured work teams having dedicated personnel with specific expertise (including bioengineering, statisticians, and skilled neurophysiologists), advanced neurophysiological techniques, not negligible economical resources, and extended cultural background and technical skills, which are not always easily implemented. In addition, it will be necessary to demonstrate further the clinical utility of functional approach in single-subject analyses. Moreover, the automatization and simplification of such complex analysis tools will be necessary to enable rapid, robust, and reproducible interpretation of data at the point of care for clinical decision-making. This will also help to further demonstrate and clarify the role of aberrant network connectivity in the neurobiology of impaired consciousness including cortical and subcortical networks interaction to generate and recover awareness.

## 4. Conclusions

In conclusion, we may argue that electrophysiologically-based diagnostic procedures represent an important resource for diagnosis, prognosis, and, therefore, management of patients with DoC, using advance passive and resting state paradigm analyses for the patients who lie in the “greyzones” between MCS, UWS, and fLIS. Furthermore, such approaches may also allow going from diagnostic stimulation to therapeutic neuromodulation [79]. Of course, it is not always possible to achieve a satisfactory diagnostic/prognostic outcome due to their experimental nature and the limitations in applicability/reliability. Such paraclinical approaches result anyway useful to better understand DoC pathophysiology, which will be important in the future to more properly diagnose and manage such very frail and vulnerable patients.

## Figures and Tables

**Table 1 brainsci-10-00042-t001:** Survey of the different neurophysiological techniques of stimulation to investigate brain function in UWS and MCS.

Technique	Availability	Ease of Application	Analysis Complexity	Information on Brain Connectivity	Diagnostic Utility
resting state EEG	high	high	moderate-high	significant	allows differentiating UWS/MCS but not identifying fLIS unless using advanced analyses (dWPLI, graph theoretic network lagged-phase synchronization, network parameters)
short-latency EPs	high	high	low	low	do not allow clear UWS/MCS differentiation but can be useful concerning prognosis
long-latency EPs (ERPs)	high	high	moderate-high	moderate	allows differentiating UWS/MCS but not identifying fLIS unless using advanced analyses (lagged-phase synchronization and network parameters following NIBS) or dedicated stimulation approaches (e.g., VMI and AMI)
TMS-EEG	low	low	high	significant	allows differentiating UWS/MCS and identifying fLIS by using advanced analyses
sleep assessment	moderate	moderate	moderate	moderate	is more useful concerning prognosis than differential diagnosis, as sleep patterns are significantly related to outcome
pain assessment	low-moderate	moderate	moderate	moderate	allows differentiating UWS/MCS and identifying fLIS unless assessing the cognitive components of the evoked responses and using advanced analyses (e.g., LEPs single features, GBO, response to TMS)

Legend: AMI, audio-motor integration; dWPLI, directed Weighted Phase Lag Index; EEG, electroencephalogram; EPs, evoked potentials; ERPs, event-related potentials; fLIS, functional Locked-In Syndrome; GBO, gamma-band oscillations; LEPs, laser-evoked potentials; MCS, minimally conscious state; NIBS, non-invasive brain stimulation; TMS, transcranial magnetic stimulation; UWS, unresponsive wakefulness syndrome; VMI, visuo-motor integration.

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
