# Peer review of "Toward Improving Diagnostic Strategies in Chronic Disorders of Consciousness: An Overview on the (Re-)Emergent Role of Neurophysiology"

_brainsci, 2020, doi:10.3390/brainsci10010042_

Round 1
Reviewer 1 Report
This manuscript presents an overview of the use of neuroimaging (fMRI, PET, but primarily EEG) to make differential diagnoses regarding disorders of consciousness. While this presents a nice overview of the various technologies/methods available, more synthesis of the results is needed.
First and most importantly, more information is needed on how the review was conducted. Was this a systematic review? A lot of the papers are from the author’s group. Is this an overview of the author’s work with DOC or is this more general? Please be specific. How were articles included? What search terms were used?
Please rephrase line 85 “without requiring collaboration with the patient” as this makes it sound a bit unethical. A statement regarding informed consent and ethics with these patients should be included. How do you get informed consent with this population? How can we ensure we behave ethically with patients who cannot respond?
The lack of figures and tables is a weakness of the manuscript. I would suggest adding a figure showing UWS v MCS vs fLIS using one or more of the proposed methods or a table outlining the various methods and what they are used to differentiate (UWS v MCS? MCS v fLIS?).
fMRI, PET and other imaging methods are usually used to study group differences. What is the sensitivity/specificity when it comes to looking at a single individual’s data relative to a group? Can neural activity to a bright light or one’s name be accurately measured at an individual level? How easy is it for a clinician (not a researcher) to analyze and interpret these data? This is mentioned briefly in the conclusions but should be expanded upon as these issues are critical to implementation.
The authors suggest "matching pain and sleep data" to "improve DoC differential diagnosis." More conclusions like this are needed. What are your suggestions for the best practice for DoC differential diagnosis? Should we combine active and passive EEG and fMRI connectivity tests on everyone? What about cost considerations? Is there some low cost test like EEG you could use to diagnose a certain % and then only require more expensive PET and fMRI for a subset?
Author Response
Dear Editor,
We want to thank you and your reviewer for the appreciation for our manuscript and for the useful comments to improve its quality.
First and most importantly, more information is needed on how the review was conducted. Was this a systematic review? A lot of the papers are from the author’s group. Is this an overview of the author’s work with DOC or is this more general? Please be specific. How were articles included? What search terms were used?We have now better specified that ours and other research groups made many efforts for identifying putative awareness markers in both resting-state and passive paradigms, mainly adopting advanced EEG/ERP analyses. We therefore proposed a scoping review (that identifies nature and extent of research evidence, including the ongoing research) aimed at highlighting a (re-)emergent role of neurophysiology in terms of current, innovative neurophysiological examination methods in resting state/passive modality, so to improve the diagnostic strategies in chronic DOC in support of the irreplaceable value of bedside, prolonged, clinical examination.
Please rephrase line 85 “without requiring collaboration with the patient” as this makes it sound a bit unethical. A statement regarding informed consent and ethics with these patients should be included. How do you get informed consent with this population? How can we ensure we behave ethically with patients who cannot respond?We rewritten the sentence since misleading. We indeed referred to experimental tasks in which the subjects is not required to actively participate but he/she is provided with stimuli that can activate specific neural pathways underpinning cognitive processes related to awareness. As required, we always obtained the written informed consent from patients’ surrogate to participate to each of the study we carried out.
The lack of figures and tables is a weakness of the manuscript. I would suggest adding a figure showing UWS v MCS vs fLIS using one or more of the proposed methods or a table outlining the various methods and what they are used to differentiate (UWS v MCS? MCS v fLIS?).Accordingly, we added a table outlining the various methods and what they are used to differentiate (i.e. covert awareness brain function in patients with UWS –fLIS– vs. UWS and MCS).
fMRI, PET and other imaging methods are usually used to study group differences. What is the sensitivity/specificity when it comes to looking at a single individual’s data relative to a group? Can neural activity to a bright light or one’s name be accurately measured at an individual level? How easy is it for a clinician (not a researcher) to analyze and interpret these data? This is mentioned briefly in the conclusions but should be expanded upon as these issues are critical to implementation.Accordingly, we expanded and better highlighted (also by means of a table) in each sub-paragraph the issues of sensitivity/specificity, availability and interpretability of the various neurophysiological methods reviewed here, as usefully suggested by the reviewer, by outlining the various methods and what they are used to differentiate (i.e. covert awareness brain function in patients with UWS –fLIS– vs. UWS and MCS).
The authors suggest "matching pain and sleep data" to "improve DoC differential diagnosis." More conclusions like this are needed. What are your suggestions for the best practice for DoC differential diagnosis? Should we combine active and passive EEG and fMRI connectivity tests on everyone? What about cost considerations? Is there some low cost test like EEG you could use to diagnose a certain % and then only require more expensive PET and fMRI for a subset?We thank the reviewer for all these stimulating questions, which were very useful to enrich the discussion on the role of neurophysiology in terms of current, innovative neurophysiological examination methods in resting state/passive modality, so to improve the diagnostic strategies in chronic DOC. We further expanded the discussion by better highlighting the different neurophysiological approaches that can be profitably "enhanced" in the study of DoC patients with their pros and cons.
Kindest regards,
The authors
Reviewer 2 Report
Comments to authors
Authors focused their paper on re-emergence of neurophysiological assessment of DOC patients in order to improve diagnosis between UWS, MC and LIS, a topic really intriguing.
As a clinical neurophysiologist, I am very supportive of the works that highlight the diagnostic possibilities of the neurophysiology, even in the challenging context of DOC. The work appears well structured and supported by previous publications of the same group and I endorse (some of) the conclusions of the authors.
However the manuscript should be enriched by the authors.
Major criticisms
Within work, the importance of sleep appears to be underestimated. In the last decade, many works support a relevant role of sleep evaluation as a parameter useful to decide further rehabilitative work-up in chronic DOC patients, given that clinical evaluation does not always detect spare capacity. Several groups are working on this topic, and some works must certainly be included in the treatment because they further underline the usefulness of a PSG in the evaluation of DOC patients. As examples, in Rossi Sebastiano et al., clin neurophysiol 2017, authors demonstrated that UWS patients with even severe impairment show residual sleep patterns (included, sometimes, REM sleep), and that the presence of slow-wave sleep is correlated with high CRS-R scores. Again, Arnaldi et al., clin neurophysiol 2016, showed that “regular” sleep structure is a good predictor of clinical outcome in sub-acute DOC patients and it is stronger than other known prognostic factors of DOC outcome. These two works highlighted that simple and (relatively) easy visual-quantitative sleep analysis is helpful in the prognostic and/or diagnostic evaluation of DOCs. Furthermore, it would be interesting if authors could mention also the recent, interesting work made by Rosanova et al., nat commun 2018, in which authors showed that loss of brain complexity after severe injuries is due to a pathological tendency of cortical circuits to fall into silence (OFF-period) upon receiving an input, a behavior typically observed during sleep. Sometimes in some sentences of the manuscript, there is a shifting from past simple to past perfect tense of the verbs. Please check the concordance of the verbs in the manuscript.
Minor revisions
Introduction, line 61; I am not sure that all the readers know the difference between locked-in syndrome and functional locked-in syndrome and “fLIS” appears for the first time in the manuscript. Hence, please add the full name here in the text e Introduction, lines 81-82; “PET”, “fMRI”, “EEG” and “MEG” appear for the first time in the manuscript. Please add the full name here in the text (for EEG, the word “Electroencephalogram” could be deleted in EEG assessment, line 131, literature review). Introduction, lines 120, “limited signs of lucidity”. I understand the meaning of “lucidity”, even if it is not a medical term. I would prefer that authors change this term (awareness?) EEG assessment, lines 136-137, “studies have shown a greater presence of delta activity in UWS then MCS”; the sentence is not explained very well, you probably wanted to underline that in UWS patients there is a greater relative power in the delta band compared to MCS. Please add the reference EEG assessment, lines 138, “deceleration of brain activity”; In my opinion the term deceleration is incorrect. Please change it with “slowing”. Evoked Potentials, line166, “ERs”, please change ERs in ERPs. Evoked Potentials, line169-171, “the absence of brainstem response in SEPs and BAEPs is indicative of poor prognosis..”, probably you mean that cortical response at SEPs is indicative of poor prognosis, because the absence of N20 is related to a poorer prognosis in comatose patients. Please change the sentence. Evoked Potentials, line 177, “NIBS” appears for the first time in the manuscript. Please add the full name here in the text. Pain, line 220, “GBO” appears for the first time in the manuscript. Please add the full name here in the text. Sleep, line 230, “REM” and NREM” appear for the first time in the manuscript. Please add the full name here in the text. Line 269, “tDCS” appears for the first time in the manuscript. Please add the full name here in the text.
Personal consideration
I do not completely agree with the last paragraph of the "authors point of view and conclusions" (lines 320-332). From my point of view, there are two different neurophysiological approaches that can be profitably "enhanced" in the near future in the study of DOC patients:
A) from a strictly clinical use, simple and very consistent neurophysiological tests (such as the presence of cortical responses to SSEPs, the presence of P300 and MMN to ERPs, the presence of an "EEG" well-structured both in wakefulness and in sleep) should be used in intensive care units and first level rehabilitation, in order to help in the differential diagnosis and outcome prognosis of patients; B) "advanced" and innovative neurophysiological investigations must be used in second and third level institutes, mainly for research purposes or, possibly, as a second diagnostic step only in restricted and selected groups of DOC patients, namely patients which lie in the “greyzones” between MCS and UWS or MCS and LIS.
The most recent studies, while being very interesting and very important for clarifying the pathophysiolological and plastic process in the disorder of consciousness, are virtually "unusable" in smaller centers, because they underlie the presence of very well structured work teams, advanced neurophysiological techniques and “extended” cultural background and technical skills which are not always easily implemented. Obviously this is a my personal consideration and I don't want to force authors to put it in their manuscript, but I would like they express their considerations about this point of view.
Author Response
Dear Editor,
We want to thank you and your reviewer for the appreciation for our manuscript and for the useful comments to improve its quality.
Within work, the importance of sleep appears to be underestimated. In the last decade, many works support a relevant role of sleep evaluation as a parameter useful to decide further rehabilitative work-up in chronic DOC patients, given that clinical evaluation does not always detect spare capacity. Several groups are working on this topic, and some works must certainly be included in the treatment because they further underline the usefulness of a PSG in the evaluation of DOC patients. As examples, in Rossi Sebastiano et al., clin neurophysiol 2017, authors demonstrated that UWS patients with even severe impairment show residual sleep patterns (included, sometimes, REM sleep), and that the presence of slow-wave sleep is correlated with high CRS-R scores. Again, Arnaldi et al., clin neurophysiol 2016, showed that “regular” sleep structure is a good predictor of clinical outcome in sub-acute DOC patients and it is stronger than other known prognostic factors of DOC outcome. These two works highlighted that simple and (relatively) easy visual-quantitative sleep analysis is helpful in the prognostic and/or diagnostic evaluation of DOCs. Furthermore, it would be interesting if authors could mention also the recent, interesting work made by Rosanova et al., nat commun 2018, in which authors showed that loss of brain complexity after severe injuries is due to a pathological tendency of cortical circuits to fall into silence (OFF-period) upon receiving an input, a behavior typically observed during sleep.
We are grateful to the reviewer for all these interesting cues on sleep topic, which were all of them adopted as very useful to enrich the paragraph in question.
Sometimes in some sentences of the manuscript, there is a shifting from past simple to past perfect tense of the verbs. Please check the concordance of the verbs in the manuscript.Checked and corrected
Introduction, line 61; I am not sure that all the readers know the difference between locked-in syndrome and functional locked-in syndrome and “fLIS” appears for the first time in the manuscript. Hence, please add the full name here in the text e IntroductionCorrected.
lines 81-82; “PET”, “fMRI”, “EEG” and “MEG” appear for the first time in the manuscript. Please add the full name here in the text (for EEG, the word “Electroencephalogram” could be deleted in EEG assessment, line 131, literature review).Corrected.
Introduction, lines 120, “limited signs of lucidity”. I understand the meaning of “lucidity”, even if it is not a medical term. I would prefer that authors change this term (awareness?).Corrected.
EEG assessment, lines 136-137, “studies have shown a greater presence of delta activity in UWS then MCS”; the sentence is not explained very well, you probably wanted to underline that in UWS patients there is a greater relative power in the delta band compared to MCS. Please add the referenceChecked and corrected.
EEG assessment, lines 138, “deceleration of brain activity”; In my opinion the term deceleration is incorrect. Please change it with “slowing”.Corrected.
Evoked Potentials, line166, “ERs”, please change ERs in ERPs.Corrected.
Evoked Potentials, line169-171, “the absence of brainstem response in SEPs and BAEPs is indicative of poor prognosis..”, probably you mean that cortical response at SEPs is indicative of poor prognosis, because the absence of N20 is related to a poorer prognosis in comatose patients. Please change the sentence.Corrected as suggested.
Evoked Potentials, line 177, “NIBS” appears for the first time in the manuscript. Please add the full name here in the text.Corrected.
Pain, line 220, “GBO” appears for the first time in the manuscript. Please add the full name here in the text.Corrected.
Sleep, line 230, “REM” and NREM” appear for the first time in the manuscript. Please add the full name here in the text.Corrected.
Line 269, “tDCS” appears for the first time in the manuscript. Please add the full name here in the text.Corrected.
Personal consideration. I do not completely agree with the last paragraph of the "authors point of view and conclusions" (lines 320-332). From my point of view, there are two different neurophysiological approaches that can be profitably "enhanced" in the near future in the study of DOC patients: A) from a strictly clinical use, simple and very consistent neurophysiological tests (such as the presence of cortical responses to SSEPs, the presence of P300 and MMN to ERPs, the presence of an "EEG" well-structured both in wakefulness and in sleep) should be used in intensive care units and first level rehabilitation, in order to help in the differential diagnosis and outcome prognosis of patients; B) "advanced" and innovative neurophysiological investigations must be used in second and third level institutes, mainly for research purposes or, possibly, as a second diagnostic step only in restricted and selected groups of DOC patients, namely patients which lie in the “greyzones” between MCS and UWS or MCS and LIS. The most recent studies, while being very interesting and very important for clarifying the pathophysiolological and plastic process in the disorder of consciousness, are virtually "unusable" in smaller centers, because they underlie the presence of very well structured work teams, advanced neurophysiological techniques and “extended” cultural background and technical skills which are not always easily implemented. Obviously this is a my personal consideration and I don't want to force authors to put it in their manuscript, but I would like they express their considerations about this point of view.We read with great interest your personal consideration and we really appreciated it. We like the way you summarized the different neurophysiological approaches that can be used to differentiate DoC patients. Therefore, we adopted you suggestion to better discuss our review findings. Thanks again for this very useful suggestion.
Kindest regards,
The authors
Round 2
Reviewer 1 Report
The authors were very responsive to my comments and the manuscript is much improved. I have no additional revisions.